# Impact of an Oral Nutritional Protocol with Oligomeric Enteral Nutrition on the Quality of Life of Patients with Oncology Treatment-Related Diarrhea

**DOI:** 10.3390/nu13010084

**Published:** 2020-12-29

**Authors:** Alejandro Sanz-Paris, Javier Martinez-Trufero, Julio Lambea-Sorrosal, Raimon Milà-Villarroel, Fernando Calvo-Gracia

**Affiliations:** 1Department of Endocrinology and Nutrition, Miguel Servet Hospital, 50009 Zaragoza, Spain; 2Instituto de Investigación Sanitaria Aragón (IIS Aragon), 50009 Zaragoza, Spain; 3Department of Medical Oncology, Miguel Servet Hospital, 50009 Zaragoza, Spain; jmtrufero@seom.org; 4Department of Oncology, University Clinic Hospital, 50009 Zaragoza, Spain; juliolambea@yahoo.es; 5Group Research on Wellbeing (GRoW), Blanquerna School of Health Sciences-Universitat Ramon Llull, 08025 Barcelona, Spain; raimonmv@blanquerna.url.edu; 6Department of Endocrinology and Nutrition, University Clinic Hospital, 50009 Zaragoza, Spain; fcalvo@comz.org

**Keywords:** oligomeric enteral nutrition, quality of life, oncology treatment-related diarrhea, nutritional protocol

## Abstract

(1) Background: Nutritional status can influence the quality of life (QoL) of cancer patients. (2) Methods: This subanalysis evaluated the impact of an oral oligomeric enteral nutrition (OEN) protocol on the QoL of patients with oncology treatment-related diarrhea (OTRD) in a multicenter, observational, prospective study (DIAPOENO study). QoL was assessed with the Nottingham Health Profile (NHP) at baseline and after eight weeks of OEN treatment. (3) In the overall population, all the NHP categories significantly improved after eight weeks of OEN treatment: energy levels (*p* < 0.001), pain (*p* < 0.001), emotional reactions (*p* < 0.001), sleep (*p* < 0.001), social isolation (*p* = 0.023), and physical abilities (*p* = 0.001). QoL improvement was higher in patients with improved or maintained nutritional status and in those with improved consistency of stools with the OEN protocol. However, QoL did not significantly improve in patients with worse nutritional status and with worse or maintained stool consistency with the OEN protocol. QoL improved regardless of disease severity. Multivariate logistic regression analysis showed that weight change was significantly associated with improved QoL (OR 2.90–5.3), except for social isolation, in models unadjusted and adjusted to age, sex, oncology treatment, and stool consistency. (4) Conclusion: In this subanalysis, the OEN protocol was associated with improved QoL.

## 1. Introduction

Cancer patients frequently suffer from gastrointestinal symptoms that severely impair their nutritional status [1]. Nutritional status can be further compromised by anticancer therapy side effects, including taste changes, nausea, constipation, and diarrhea [2]. Oncology treatment-related diarrhea (OTRD) is a common side effect [3], resulting in low performance, frequent hospital admissions, reduced survival, and impaired quality of life (QoL) [4]. The development of targeted therapies that are often maintained for prolonged periods has increased the frequency of OTRD [5].

QoL reflects people’s wellbeing by considering emotional, social, and physical aspects of life [6]. This multidimensional measure is key in oncology, given the impairment that cancer and oncology treatments promote in physical and psychological spheres of life [7]. Common psychological consequences of malnutrition and diarrhea are anxiety and depression, which can severely restrict daily life activities [8]. Moreover, the higher incidence of hospital admissions, longer hospital stays, and reduced tolerance to anticancer treatments associated with malnutrition and diarrhea result in poorer outcomes, reduced QoL, and shorter survival [9,10]. A systematic review identified that diarrhea episodes correlated with the lowest QoL perception in cancer patients [7].

Given the impact of nutrition on disease outcomes and QoL, nutritional support is expected to improve the wellbeing of cancer patients by reducing weight loss and other gastrointestinal complications, shortening the length of hospital stays, and improving tolerance to treatments [11]. However, the association between QoL and nutrition in cancer patients has been scarcely studied [12].

A systematic review showed that oral nutritional support benefited some aspects of QoL (emotional functioning, dyspnea, loss of appetite, and global QoL) [13]. A more recent systematic review only found significant differences in some QoL domains with high-protein, n-3 polyunsaturated fatty-acid-enriched oral nutritional supplements [9]. However, both systematic reviews were limited by the reduced number of studies included and their heterogeneity.

Nutritional support comprises dietary counseling, food supplements, and nutrition therapy [14]. However, clinical guidelines mainly provide a pharmacological approach to treat diarrhea, and the nutritional support of patients is often disregarded [3,4,15,16,17,18]. We recently published an oral oligomeric enteral nutrition (OEN)-based protocol and showed its effectiveness in patients with OTRD [19,20]. OEN is where proteins are provided as peptides instead of whole proteins and fats as medium-chain triglycerides with the aim of increasing nutrient absorption. The protocol was proven effective in improving the nutritional status and consistency of stools in patients with OTRD [20]. Since there is little evidence on the efficacy of OEN in OTRD [19,20] and, to our knowledge, no previous study has evaluated its impact on QoL in patients with cancer, this subanalysis evaluated whether the effectiveness of the OEN protocol translated into an improved QoL in patients with OTRD.

## 2. Materials and Methods

### 2.1. Study Design

This study reports the subanalysis of QoL data from the DIAPOENO study: a multicenter, observational, prospective cohort study [20]. The study was conducted at 15 centers across Spain and was approved by the Ethics Committee of Comunidad Autónoma de Aragón (CEICA, Code CP-CI.PI 15/0319). The study was performed in accordance with the tenets of the Declaration of Helsinki and written informed consent was obtained from all participants.

The design and procedures of the study have been previously described in detail [20]. In brief, patients received, at the onset of OTRD, Survimed OPD Drink^®^ (Fresenius Kabi, Bad Homburg, Germany), a nutritionally complete formula. This oral nutritional support is composed of hydrolyzed proteins (18.6%), carbohydrates (56.4), and fat content (25%). Depending on nutritional status and intestinal function, 2 or 3 bottles were administered per day (200 mL/bottle with a caloric density of 1 Kcal/mL) of oral OEN [19].

Inclusion criteria were age ≥18 years, cancer diagnosis, treatment with cancer therapy (targeted therapy, chemotherapy, radiotherapy, or chemotherapy and radiotherapy), and OTRD-related malnourishment or risk of malnourishment. Exclusion criteria included: life expectancy <3 months, diarrhea associated with antibiotics, H_2_-receptor antagonists or prokinetics, laxatives or osmotically active agents treatments, *Clostridium difficile* infection, and the presence of other gastrointestinal conditions that could interfere with the study assessments [20].

### 2.2. Study Outcomes

The main objective of this subanalysis was to evaluate the impact of the OEN protocol on the QoL of patients. The secondary objectives of the study were: to assess whether the improvement in QoL correlated with nutritional status, disease severity, and consistency of stools; and to identify independent factors significantly associated with the improvement of QoL categories.

QoL was assessed using the Nottingham Health Profile (NHP). The NHP is a multidimensional measure of health comprising a first part with 38 questions across 6 subareas: energy levels (*n* = 3), pain (*n* = 8), emotional reactions (*n* = 9), sleep (*n* = 5), social isolation (*n* = 5), and physical abilities (*n* = 8). Statements are assigned a weighted value using the Thurstone method of paired comparisons, giving a score that can range from 0 (minimum severity) to 100 (maximum severity) [21].

The NHP was administered at baseline and following 8 weeks of OEN treatment. In this study, the Spanish-validated version of the questionnaire was employed under the user license of the copyrighted material [22].

QoL change was also assessed in patient subgroups based on nutritional status, disease severity, and stool consistency. The protocol was deemed effective in the previous study when nutritional status was maintained at risk of malnourishment or improved 1 or 2 levels (from moderate or severe malnourishment). Worsening of nutritional status was considered when: nutritional status worsened 1 (from at risk of malnourishment or from moderate malnourishment) or 2 levels (from at risk of malnourishment).

The protocol was considered effective when stool consistency, evaluated with the Bristol Stool form scale [16], was maintained at normal stool types or improved by 1, 2, or 3 levels. Stool consistency worsened when stool classification increased by 1 or 2 levels [20]. In this subanalysis, QoL changes were assessed in patients with improved, maintained, or worsened nutritional status with the OEN protocol, with improved, maintained, or worsened stool consistency with the OEN protocol and in patients with severe (presence of metastasis or palliative treatment) or nonsevere disease.

### 2.3. Statistical Analyses

Distribution of data distribution was assessed with the Kolmogorov–Shapiro–Wilks test and normality plots. Continuous variables were described as mean and standard deviations and categorical variables as numbers and percentages. Differences in QoL from baseline to Week 8 were estimated using the Wilcoxon paired test.

Logistic regression models were performed to determine factors independently associated with the improvement in QoL categories. The independent effects of sex, age, palliative treatment, weight change, and stool consistency improvement on QoL were assessed, and results were expressed as odds ratios (OR) with 95% CI. The effect of weight change was assessed in adjusted and unadjusted models. Analyses were performed with the statistical package IBM SPSS Statistics v.24.0 (IBM Corp, Armonk, NY, USA). Statistical significance was set at *p* < 0.05.

## 3. Results

### 3.1. Study Population

Out of a total of 162 patients with OTRD, 149 completed the study, and QoL data were available for 135. In the overall population, mean ± SD age was 68.6 ± 12.6 years, and 55% were men and no patient received previous enteral nutrition [20]. Table 1 shows baseline characteristics categorized by disease severity. Severe disease was considered for patients with metastases or on palliative treatment.

### 3.2. QoL Improvement

At baseline, energy levels (38.97) and sleep (35.47) scored higher (worse perceived QoL) than other categories, whereas social isolation (14.34) and pain (16.15) showed lower scores (better perceived QoL).

After eight weeks of OEN treatment, all the NHP categories significantly improved: energy levels (*p* < 0.001), pain (*p* < 0.001), emotional reactions (*p* < 0.001), sleep (*p* < 0.001), social isolation (*p* = 0.023), and physical abilities (*p* = 0.001). Differences after eight weeks of OEN treatment were more pronounced for energy levels (difference of 13.12) and emotional reactions (difference of 10.38). Lower differences after eight weeks of OEN treatment were observed for social isolation (difference of 3.84) and sleep (difference of 5.81) (Figure 1).

We next assessed the subgroup of patients that better improved their QoL. To this end, QoL scores were analyzed according to nutritional status, disease severity, and consistency of stools. After eight weeks of OEN treatment, all the NHP categories were improved, with significantly higher differences in those who improved or maintained nutritional status vs. those with worsened nutritional status for energy levels (*p* = 0.002, *p* = 0.001), pain (*p* = 0.002, *p* = 0.001), emotional reactions (*p* = 0.001, *p* = 0.001), sleep (*p* = 0.036, *p* = 0.021), and physical abilities (*p* = 0.001, *p* = 0.026). In contrast, social isolation showed significant differences in patients with worsened nutritional status (*p* = 0.024) (Figure 2).

Disease severity was not associated with significant differences in QoL improvement. Patients with severe disease significantly improved QoL scores for energy levels (*p* = 0.001), pain (*p* = 0.030), emotional reactions (*p* = 0.001), and sleep (*p* = 0.044). Similarly, the subgroup of patients with nonsevere disease significantly improved energy levels (*p* = 0.001), pain (*p* = 0.001), emotional reactions (*p* = 0.001), and sleep (*p* = 0.002). For social isolation and physical abilities, differences were not significant in patients with severe and nonsevere disease (*p* = 0.124, *p* = 0.070), respectively (Figure 3).

The improvement in QoL categories was higher in patients with improved consistency of stools from baseline to Week 8, and significant for energy levels (*p* = 0.001), pain (*p* = 0.001), emotional reactions (*p* = 0.001), sleep (*p* = 0.003), social isolation (*p* = 0.003), and physical abilities (*p* = 0.001). In patients with maintained or worsened consistency of stools, significant differences from baseline to Week 8 were found only in the perception of pain (*p* = 0.044, *p* = 0.039) (Figure 4).

Multivariate logistic regression analysis was performed to determine independent factors significantly associated with the improvement in QoL. The following variables were associated with an improved perception of energy levels: women (OR, 0.39; 95% CI, 0.15–0.998), palliative treatment (OR, 0.26; 95% CI, 0.10–0.64), and weight change (OR, 4.90; 95% CI, 1.83–9.03). The improvement in pain perception was associated with weight change (OR, 3.79; 95% CI, 1.42–1.42) and improvement in stool consistency (OR, 3.76; 95% CI, 1.38–10.56). Weight change was the only determinant significantly associated with improved emotional reactions (OR, 5.30; 95% CI, 2.1–2.05) and sleep (OR, 4.19; 95% CI, 1.57–11.20). Patients under palliative care were less likely to improve social isolation perception (OR, 0.29; 95% CI, 1.0–0.88). The improvement in physical abilities was significantly associated with age (OR, 1.04; 95% CI, 1.00–1.08), palliative treatment (OR, 0.14; 95% CI, 0.06–0.35), weight change (OR, 4.95; 95% CI, 1.92–12.79) and improved stool consistency (OR, 3.74; 95% CI, 1.40–12.50). Since weight change was an independent factor of improvement for all the categories but social isolation, we calculated the effect of weight change in adjusted and unadjusted models (Table 2).

## 4. Discussion

This subanalysis showed the overall improvement in the QoL of patients treated with the previously described OEN protocol [20]. This improvement agrees with the main findings of the DIAPOENO study, showing that the OEN protocol was effective in improving nutritional status and consistency of stools in patients with OTRD. In this subanalysis, we reveal that the effectiveness of the OEN protocol translates into QoL improvement, with significant differences in all the NHP categories in the overall population. The improvement in QoL was more pronounced in patients with improved or maintained nutritional status and in those with improved consistency of stools but was independent of the severity of the disease.

Studies addressing the impact of nutritional support on the wellbeing of cancer patients are scarce, although a study revealed that global QoL scores are significant predictors of survival in lung cancer patients [23]. Moreover, systematic reviews include heterogeneous studies with inconclusive results, making it difficult to obtain reliable conclusions. Whereas some studies found a positive effect of nutritional support on patients’ QoL [9,13,24], others did not [25,26].

We observed that at baseline, the most compromised NHP categories were energy levels and sleep, whereas social isolation and pain were better perceived. This agrees with previous studies in patients treated with radiotherapy for oropharyngeal or epipharyngeal cancer [27] and with nonfunctioning pituitary adenomas [28], showing higher scores for energy levels and lower for pain and social isolation.

After receiving the OEN protocol, patients significantly improved the perception of all the NHP categories, indicating an overall improved QoL upon this nutritional support. Social isolation, physical abilities, and sleep were the categories with more subtle differences from baseline. Of note, social isolation was not severely compromised at baseline (lowest score among all the categories), which could indicate that this category is not representative of the distress caused by malnutrition in OTRD patients. Regarding sleep, although patients showed a negative perception at baseline, it was among the categories with lower differences from baseline. A possible explanation could be that sleep affectation is intrinsically related to cancer itself and does not depend on the nutritional status of patients. However, this assumption requires further confirmation. A previous multivariate logistic regression analysis found that sleep disorders were strongly associated with low QoL and the authors pointed to probable undertreatment of the condition by physicians [29].

The most remarkable improvement was observed in energy levels (difference of 13.12) and emotional reactions (difference of 10.38). Such an improvement in energy levels agrees with the positive effect of the OEN protocol on nutritional status [20]. Low energy and fatigue are commonly observed in cancer patients receiving therapy. This is because the metabolism of cancer patients is altered, with higher proteolysis and lipolysis, increased hepatic production, reduced insulin sensitivity, and, therefore, increased energy expenditure. In this context, nutritional support can help maintain the balance between energy expenditure and food intake, increasing energy levels [30]. The improvement in emotional reactions could also be the consequence of the improvement in the consistency of stools previously reported [20], as OTRD is associated with anxiety and depression, which, in turn, can compromise daily life activities [7].

One of the most interesting results of this study is that the improvement in QoL was significantly higher in patients showing an improvement or maintenance of nutritional status with the OEN protocol and in those with improved consistency of stools. However, it was observed regardless of disease severity. These results suggest that the positive effect of the OEN protocol on nutritional status and consistency of stools influences the QoL of patients independently of disease severity.

The scores of NHP categories were higher at baseline (more severe QoL impairment) for patients in our study compared to the general population in a subgroup of 1220 Spanish participants [22]. After the OEN treatment, however, the score of emotional reactions was lower than that reported in the general population aged 60–69 and >70 years. Sleep, social isolation, and physical abilities scores were also lower compared with the population aged >70 years [22]. This evidences the level of improvement associated with the OEN protocol, with QoL scores after eight weeks being close to the general population.

We next analyzed the effects of sex, age, palliative treatment, weight changes, and stool consistency in QoL improvement by logistic regression models. Improved perception of energy levels was associated with male sex, curative treatment, and weight change. Although QoL improvement depends on the evolution of stool consistency in bivariate models, stool consistency only correlated with pain and physical abilities improvement in the multivariate model. This result is surprising as OTRD can result in increased levels of anxiety, depression, and restrict daily life activities [8].

Increasing age was a positive determinant of physical abilities and the palliative treatment of worsened energy levels and social isolation. The negative effect of palliative treatment on QoL is consistent with the results of our previous study [20], showing poorer nutritional status and compliance rates (80% of patients under curative care consumed the total OEN content vs. 58.5% of patients under palliative care) in patients under this treatment modality.

The most important determinant of QoL improvement in our analysis was weight change, being an independent factor of improvement for all the categories but social isolation. The effect of weight on QoL has been previously reported, with weight loss at presentation being associated with poorer survival and QoL [13], worse long-term prognosis [31], and nonrelapse mortality in cases of severe weight loss (>10%) [32]. Remarkably, patients treated with OEN showed a significant increase in BMI (−0.29; *p* = 0.004) and weight (−0.77; *p* = 0.007) after eight weeks in our previous study [20], which could help explain the improvement in QoL observed in this subanalysis. Unlike other clinical variables, weight is a noticeable feature for the patient, likely having an important effect on QoL.

Different determinants of QoL were found in previous studies. In a prospective cross-sectional study conducted in 271 head and neck, esophagus, stomach, and colorectal cancer patients, QoL scores correlated with cancer location, nutritional intake, weight loss, chemotherapy, surgery, disease duration, and stage of disease [24]. In these patients, malnutrition was associated with poorer global QoL and reduced physical, role, cognitive, emotional, and social QoL scales [24]. The authors identified the stage of the disease as the major determinant of QoL followed by nutritional status and dietary intake [24]. In a Phase II trial of patients with nonsquamous, non-small-cell lung cancer (NSCLC), variations in QoL scores were determined by treatment cycles [33]. In a study including 772 cancer patients, impaired appetite, more symptoms, presence of metastasis, being female, and being of higher economic classes significantly impaired the QoL of patients [34].

The main limitation of this study is the lack of a control group, which is particularly relevant in a subjective assessment such as QoL. Moreover, other factors besides the OEN protocol, such as following an astringent diet by all the patients, the use of transient antidiarrheal drugs in patients with watery stools, the time from cancer treatment finalization, or the presence of anemia could have conditioned QoL results. Some acknowledged limitations of the NHP scale are that, since it is a generic tool, some of the categories could not be representative of the disease [35], and it represents severe QoL status; thus, milder forms of distress can be overlooked [21].

Despite these limitations, this study also has several strengths. As previously indicated, studies addressing the impact of nutritional support on QoL are scarce and often show inconclusive and heterogeneous results [9,13,24,25,26]. To our knowledge, no previous study evaluated the impact of OEN on the QoL of patients with OTRD. The association between improved nutritional status and QoL observed in our study is of great interest because, given the multifactorial nature of QoL, it is not common to observe a clear effect on this variable. Although weight increase is a frequent outcome in nutritional support protocols, this variable is often disregarded in several studies at the expense of other outcomes such as hospital stay or mortality. However, here we show that weight is the most important factor of QoL improvement in multivariate models, suggesting the interest in promoting interventions focusing on preventing weight loss in cancer patients. Regarding QoL measurement, we selected the NHP because there is a validated Spanish translation, which previously showed similar construct validity, accuracy, and sensitivity [22]. Moreover, the NHP is a valid and easy-to-complete questionnaire (10–15 min) that is sensitive to assess the change with time upon different interventions [21].

## 5. Conclusions

In this subanalysis of the DIAPOENO study, the OEN protocol was associated with an overall QoL improvement, with significant differences in energy levels, pain, emotional reactions, and physical abilities. The improvement in QoL was more pronounced in patients with improved or maintained nutritional status and in those with improved consistency of stools but was independent of the severity of the disease. Weight change was the most important factor of QoL improvement in multivariate models.

## Figures and Tables

**Figure 1 nutrients-13-00084-f001:**
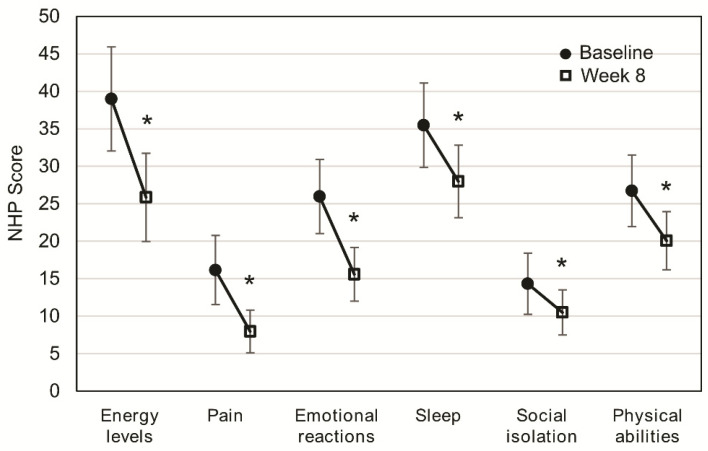
Change in quality of life after 8 weeks of oligomeric enteral nutrition (OEN). The plots show mean scores ±95% CI for each category of the Nottingham Health Profile at baseline and after 8 weeks of OEN treatment. * indicates *p* < 0.05 (from baseline to Week 8). The Nottingham Health Profile (NHP) score ranges from 0 (does not perceive any health problem) to 100 (maximum health problems perceived by the patient).

**Figure 2 nutrients-13-00084-f002:**
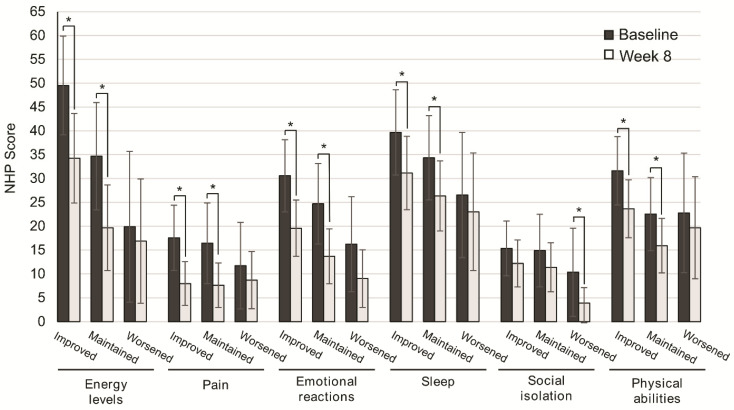
Change in quality of life after 8 weeks of oligomeric enteral nutrition (OEN) by nutritional status evolution. The bars show mean scores ±95% CI for each category of the Nottingham Health Profile at baseline and after 8 weeks of OEN treatment for patients who improved, maintained, or worsened their nutritional status with the OEN protocol. * indicates *p* < 0.05 (from baseline to Week 8). The NHP score ranges from 0 (does not perceive any health problem) to 100 (maximum health problems perceived by the patient).

**Figure 3 nutrients-13-00084-f003:**
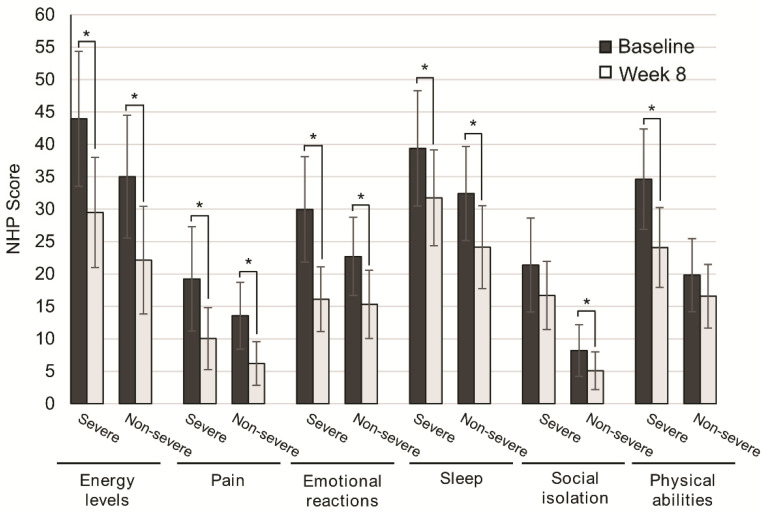
Change in quality of life after 8 weeks of oligomeric enteral nutrition (OEN) by disease severity. The bars show mean scores ±95% CI for each category of the Nottingham Health Profile at baseline and after 8 weeks of OEN treatment for patients with or without disease severity. * indicates *p* < 0.05 (from baseline to Week 8). The NHP score ranges from 0 (does not perceive any health problem) to 100 (maximum health problems perceived by the patient).

**Figure 4 nutrients-13-00084-f004:**
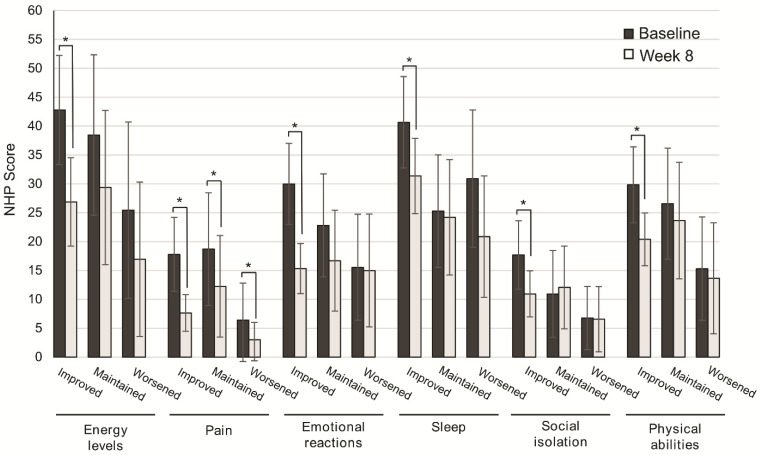
Change in quality of life after 8 weeks of oligomeric enteral nutrition (OEN) by the consistency of stool evolution. The bars show mean scores ± 95% CI for each category of the Nottingham Health Profile at baseline and after 8 weeks of OEN treatment for patients who improved, maintained, or worsened the consistency of stools with the OEN protocol. * indicates *p* < 0.05 (from baseline to week 8). The NHP score ranges from 0 (does not perceive any health problem) to 100 (maximum health problems perceived by the patient).

**Table 1 nutrients-13-00084-t001:** Demographic and clinical characteristics of study participants.

	Severe (*n* = 83)	Nonsevere (*n* = 52)	*p*-Value
Age (years), Mean (SD)	69.5 (12.8)	66.2 (12.4)	0.433
Gender, (%)			
Men	47 (56.6%)	28 (53.8%)	0.752
Women	36 (43.4%)	24 (46.2%)	
Weight (kg), Mean (SD)	60.1 (12.1)	65.4 (11.6)	0.255
BMI (kg/m^2^), Mean (SD)	21.4 (3.2)	23.6 (4.1)	0.512
Resectability (* *n* = 132)			
Unresectable	64 (80.0%)	0 (0%)	<0.001
Localized	16 (20.0%)	52 (100%)	
Type of treatment			
Palliative	62 (74.7%)	0 (0%)	<0.001
Curative	21 (25.3%)	52 (100%)	
Treatment modality			
Chemotherapy	46 (55.4%)	7 (13.5%)	<0.001
Radiotherapy	9 (10.8%)	12 (23.1%)	
Chemotherapy + Radiotherapy	28 (33.7%)	33 (63.5%)	
Targeted therapy (* *n* = 134)			
Yes	23 (27.7%)	3 (5.9%)	0.002
No	60 (72.3%)	48 (94.1%)	
Type of tumor			
Gynecologic/urologic	9 (10.8%)	14 (26.9%)	0.023
Digestive	67 (80.7%)	37 (71.2%)	
Other	7 (8.4%)	1 (1.9%)	

* indicates percentages obtained considering the number of patients with available data. BMI, Body Mass Index. Severity is defined by the presence of metastases or the type of treatment (palliative).

**Table 2 nutrients-13-00084-t002:** Multivariate logistic regression models to assess the effect of weight change on QoL improvement. The table shows the model unadjusted (raw) and adjusted to sex, age, palliative treatment, and stool consistency improvement. Data are expressed as odds ratios (OR) with 95% CI and *p*-values. Bold text indicates significant determinants.

	Energy Levels		Pain		Emotional Reactions		Sleep		Social Isolation		Physical Abilities	
Weight Change	OR (95% CI)	*p*	OR (95% CI)	*p*	OR (95% CI)	*p*	OR (95% CI)	*p*	OR (95% CI)	*p*	OR (95% CI)	*p*
OR (raw)	**3.28 (1.55** **–6.95)**	**0.002**	**2.65 (1.2** **–5.9)**	**0.017**	**4.2 (1.90–9.20)**	**<0.001**	**2.90 (1.30** **–6.50)**	**0.012**	0.96 (0.23–1.83)	0.329	**3.81 (1.79** **–8.12)**	**0.001**
OR (adjusted)	**4.90 (1.83–9.03)**	**0.000**	**3.79 (1.42–1.42)**	**0.008**	**5.30 (2.1–2.05)**	**0.001**	**4.19 (1.57–11.20)**	**0.004**	1.12 (0.39–3.27)	0.83	**4.95 (1.92–12.79)**	**0.001**

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
