# Peer review of "Impact of an Oral Nutritional Protocol with Oligomeric Enteral Nutrition on the Quality of Life of Patients with Oncology Treatment-Related Diarrhea"

_nutrients, 2020, doi:10.3390/nu13010084_

Round 1

Reviewer 1 Report

The authors present very interesting data from a subanalysis of a previous trial regarding the impact of an oligomeric enteral diet (OED) for eight weeks on quality of life in 162 cancer patients with treatment related diarrhea.  

As a matter of fact, the OED (Fresenius Survimed OPN Drink) has to be considered an oral nutritional supplement (ONS). There was obviously no enteral tube feeding. Because the ESPEN definition of  nutritional therapy (Cederholm et al, Clin Nutr 2017; 36: 49-61) differentiates between enteral nutrition (tube feeding) and ONS, this should be clarified in the title: … with an oligomeric oral nutritional supplement … 

The major limitation is the missing control group as discussed by the authors themselves. 

Further comments:

-Give the numbers of weight change more in detail

-Was there information regarding the compliance oft he patients regarding ONS intake.

-Was there during  ONS treatment any dietary counseling which might also have an influence on QoL?

-Which variables were selected for the multivariate analysis

-Discussion: …. Increased hepatic production of glucose ?

Table 2: How many patients were analyzed in the adjusted model?

-Typing errors: Line 47: mantained, Table 2: ajusted

Author Response

  What do you want to do ? New mailCopy

Reviewer 2 Report

Types de traductions

Traduction de texte

Texte source

                      1648 / 5000    

Résultats de traduction

In this article, the authors report the results of a prospective multicenter study evaluating the effects of an oligomeric enteral nutrition (OEN) protocol on the quality of life of 135 patients with oncologic treatment-related diarrhea (OTRD). This is a sub-analysis of a larger study (DIAPOENO study) that showed the impact of OEN on nutritional status and stool consistency (study published in Nutrients Journal this year). Quality of life was assessed with a validated scale (Nottingham Health Profile - NHP). All categories of NHP improved after 8 weeks of OEN, and weight change was shown to be an independent factor associated with improved quality of life. The methodology of this study is correct. The article is well written and informative.   I only have a few minor revisions: * Study population, table 1: the comparison between patients suffering from severe / non-severe illnesses is not relevant in the context of this study. * It does not seem clear to me if the patients were still under cancer treatment or not at the time of inclusion and during the 8 weeks of follow-up? If yes, the proportion of patients receiving chemotherapy / radiotherapy or targeted therapies during the relevant period should be mentioned. * The time between the end of cancer treatment and anemia are two other known factors that can affect the quality of life of cancer patients. Could the authors incorporate these factors into their analyzes? * Did the authors assess the patient's compliance with the SDO?

Author Response

  What do you want to do ? New mailCopy
